# Rotatable Bi-Channel En Bloc Resection of Bladder Tumor for Non-Muscle-Invasive Bladder Cancer in an Ex Vivo Porcine Model

**DOI:** 10.3390/cancers15174255

**Published:** 2023-08-25

**Authors:** Qiu Yao, Huizhong Jiang, Hui Niu, Guangmo Hu, Jianlong Cao, Boxin Xue

**Affiliations:** 1Department of Urology, The Second Affiliated Hospital of Soochow University, Suzhou 215004, China; 20194133099@stu.suda.edu.cn (Q.Y.); guangmo_hu@126.com (G.H.); 2Department of Operating Room, The Second Affiliated Hospital of Soochow University, Suzhou 215004, China; huizhong_jiang2023@126.com; 3Department of Pathology, The Second Affiliated Hospital of Soochow University, Suzhou 215004, China; hui_niu@yeah.net; 4Department of General Surgery, Zhangjiagang Second People’s Hospital, Zhangjiagang 215633, China; jianlongcao@126.com

**Keywords:** bladder cancer, pilot study, ex vivo model, porcine model, en bloc resection, rotatable bi-channel en bloc resection of bladder tumor, feasibility

## Abstract

**Simple Summary:**

En bloc resection of bladder tumor (ERBT) has shown promise as a treatment option for non-muscle-invasive bladder cancer (NMIBC). However, its limitation of a single working channel restricts its application. Therefore, we developed a new surgical technique called “rotatable bi-channel en bloc resection of bladder tumor (RBC-ERBT),” which allows for adjustable traction and precise surgical maneuvers. In this study, we assessed the feasibility and outcomes of RBC-ERBT using ex vivo porcine bladder models. The results demonstrated that compared with ERBT, RBC-ERBT demonstrated improved success rates, reduced resection times, and effectively managed challenging lesions. These findings highlight the potential value of RBC-ERBT as an advanced approach for treatment of NMIBC.

**Abstract:**

En bloc resection of bladder tumor (ERBT) is a promising alternative for non-muscle-invasive bladder cancer management. However, the tumor characteristics and surgeon’s experience influence its application. Therefore, in this pilot study, we developed a technique called “rotatable bi-channel en bloc resection of bladder tumor (RBC-ERBT)” and assessed its feasibility, efficacy, and safety compared with those of conventional ERBT. In an ex vivo porcine bladder model, 160 bladder lesions of varying morphologies (exophytic and flat) and sizes (1 and 2 cm) were created and evenly distributed across different locations. A total of 160 procedures were performed, with the ERBT and RBC-ERBT group each exhibiting 80 lesions. RBC-ERBT had a significantly higher technical success rate than ERBT (98.8% vs. 77.5%) for exophytic and flat lesions of both sizes and dome lesions. The procedure time was significantly shorter in the RBC-ERBT group, particularly for flat lesions, lesions with a 2 cm diameter, and lesions located at the dome. RBC-ERBT had a significantly lower piecemeal resection rate than ERBT (0% vs. 18.8%). The incidence of perforation or detrusor muscle sampling did not differ between the groups. Compared with conventional ERBT, RBC-ERBT offered improved success rates, reduced resection times, and effective management of challenging lesions.

## 1. Introduction

Bladder cancer is the 10th most common cancer worldwide [1]. Approximately 75% of patients are diagnosed with non-muscle-invasive bladder cancer (NMIBC), which is characterized by cancer limited to the urothelium or lamina propria, without detrusor muscle (DM) invasion [2]. Although transurethral resection of bladder tumor (TURBT) is the standard approach for managing NMIBC [3], resected fragments hinder accurate pathological assessment and may increase the risk of tumor cell seeding and reimplantation [4,5]. En bloc resection of bladder tumor (ERBT) has emerged as a promising alternative. This technique offers several potential advantages, such as the ability to resect the tumor and its base as a single piece and improved specimen collection for accurate pathological assessment with increased DM sampling [6,7]. Additionally, ERBT may lead to reduced perioperative complications, a decreased risk of tumor shedding and implantation, and potentially lower recurrence rates [8,9,10].

Despite the benefits of ERBT, its application can be influenced by factors such as tumor morphology, size, location, and surgeon experience. Currently, only 35% of European urologists have attempted to perform this technique whenever possible [11], and the adoption rate is even lower (2.6%) in German-speaking countries [12]. This low adoption may be attributed to surgeons’ preferences, equipment availability, and technique limitations. Due to the current limitations of the ERBT technique, as stated in the European Association of Urology guidelines [8], its feasibility is restricted to selected exophytic tumors. Moreover, approximately 30% of patients with NMIBCs are not considered suitable candidates for ERBT [13]. This highlights the challenges and limitations of ERBT. To address these limitations and expand the indications for en bloc resection, there is a need to develop novel techniques that can effectively manage lesions of different morphologies, sizes, and locations.

One limitation of conventional ERBT is that it is a single-channel procedure, regardless of whether it is performed using monopolar electrocautery, bipolar electrocautery, or laser techniques [14,15,16,17]. In these approaches, only a single instrument is used. Therefore, several new systems have been developed to overcome these limitations. For example, a telerobotic system has demonstrated the feasibility of robot-assisted ERBT in both ex vivo and in vivo studies [18,19]; however, it is not entirely a transurethral surgery, as a laparoscope needs to be inserted through the anterior bladder wall to visualize the resection process, and this system lacks depth perception. Komai et al. [20] proposed a two-arm transurethral surgical approach; however, in this system, the two arms and a single endoscope were placed in an equilateral triangular configuration, thus restricting the coordination between the two instruments.

To expand the range of indications, simplify surgical procedures, and improve surgical outcomes, we developed a novel technique called the “rotatable bi-channel en bloc resection of bladder tumor (RBC-ERBT)” in this study. The RBC-ERBT system features two independent rotatable channels, each accommodating specific instruments with different functions. One channel is used for tissue retraction and target area exposure, while the other is used for cutting or cauterization. It combines the principles of ERBT with those of a grasping assistant and utilizes a rotatable external sheath with a second working channel that has been specifically developed and patented for this purpose. This design enables more precise and controlled surgical maneuvers, potentially improving surgical outcomes.

This pilot study aimed to assess the feasibility of RBC-ERBT using an ex vivo porcine model and to compare the efficacy and safety of RBC-ERBT versus that of conventional ERBT. This study aimed to gain valuable insights into the potential benefits of RBC-ERBT and its future clinical applications in the treatment of NMIBC.

## 2. Materials and Methods

### 2.1. Study Design

The trial was a prospective ex vivo study that was exempt from institutional review board approval because it did not involve human subjects or living animals. The study was conducted at the Urological Endoscopic Training Center of the Second Affiliated Hospital of Soochow University for 8 months. Fresh bladders from adult swine were procured from a local abattoir and placed in a bladder simulator box, which was also used for ERBT training in our facility. Both the RBC-ERBT and ERBT procedures were performed by a skilled endoscopist with 8 years of experience with the ERBT technique in both human and animal models.

### 2.2. Preparation of the Porcine Bladder Lesions

In total, 160 bladder lesions were created for this study. Among them, 80 lesions were exophytic and generated by suturing the bladder mucosa (Figure 1A,D), whereas the remaining 80 lesions were flat and marked with coagulation dots (Figure 1G,J). Each type of lesion was divided equally, with half of the lesions having a diameter of 1 cm (Figure 1A,G) and the other half having a diameter of 2 cm (Figure 1D,J). The lesions were evenly distributed across the anterior, posterior, left, and right walls and the dome of the bladder, with 32 lesions at each location.

### 2.3. ERBT and RBC-ERBT

We conducted ERBT and RBC-ERBT procedures using a continuous-flow laser resectoscope (Olympus Winter & Ibe Gmbh, Hamburg, Germany) and a holmium laser system (VersaPulse PowerSuite, Lumenis, Yokneam, Israel). For RBC-ERBT, we used a specially prepared rotatable external sheath with a second working channel (Suzhou Institute of Biomedical Engineering and Technology, Suzhou, China) (Figure 2A,B).

ERBT procedures were performed according to standard ERBT protocols [21]. In RBC-ERBT, the procedure begins by creating a circumferential incision around the bladder lesion using a holmium laser, ensuring a distance of 5–10 mm from the edge of the lesion. An incision was made deep enough to visualize the DM (Figure 3A). For lesions located in areas other than the dome, the “lift-and-cut” method (Figure 3B) was employed. The base of the lesion was lifted using grasping forceps inserted through an internal working channel to expose the target area. The laser introduced through the external working channel was used to cut the tissue along an arc line. This was achieved by rotating the exterior sheath to guide resection (dashed line in Figure 3D).

For lesions situated at the dome, a “peel-off” method (Figure 3C) was utilized. Instead of lifting the tissue, forceps were used to grasp one edge of the lesion and peel the tissue toward the opposite side of the lesion. Once the target area was exposed, the fiber inserted through the other working channel was adjusted to align with the target line (indicated by the dashed line in Figure 3C), and laser cutting commenced. During the cutting process, the grip on the lesion was adjusted to maintain tension and keep the target area visible and accessible. Resection was continued until the far end of the lesion. This step constitutes the main part of the procedure and is time consuming. Clear visualization and precise movements were crucial and were aided by forceps. Additionally, by rotating the external sheath with the external channel while holding the lesion with the grasper forceps, the laser can cut the tissue along the moving curve, facilitating the dissection process.

Once the tumor was completely resected, the specimen was removed using a resectoscope. Finally, endoscopy was performed to examine the resection bed and assess changes in the bladder wall following lesion removal (Figure 3D). In cases of technical difficulty, both groups had the option of switching to laser or electrocautery piecemeal resection as required.

### 2.4. Data Collection

We documented the following parameters with the assistance of an independent observer: lesion morphology, size, and location; procedure time (in min); incidence of perforation; conversion to piecemeal resection; and overall procedural outcomes. The technical success of RBC-ERBT or ERBT was defined as complete en bloc resection using the RBC-ERBT or ERBT set without perforation or conversion to other surgical techniques. The procedure time was calculated from the initiation of the circumferential incision until specimen extraction. Perforation was defined as the visual detection of a hole in the bladder wall or occurrence of water leakage through the bladder wall during or after the procedure.

Following each ERBT and RBC-ERBT procedure, the resected specimens were carefully spread, securely placed on plastic plates (Figure 1B,E,H,K), and submerged in formalin solution. Subsequently, the specimens were sent to a pathologist to confirm the involvement of DM (Figure 1C,F,I,L).

### 2.5. Statistical Analysis

Data analysis was conducted using SPSS software (version 26.0; SPSS, IBM Company, Armonk, NY, USA). Analysis of technical success, perforation, piecemeal resection, and DM sampling events was performed using the chi-square test. Procedure time was analyzed using the Mann–Whitney U-test. Statistical significance was set at *p* < 0.05.

## 3. Results

### 3.1. Study Samples

In the ex vivo porcine models, 160 procedures were performed, 80 of which were dedicated to ERBT and 80 of which were dedicated to RBC-ERBT. Both the ERBT and RBC-ERBT groups consisted of lesions with varying morphologies (exophytic and flat) and sizes (1 and 2 cm in diameter) (Figure 4). Within each group, we evenly distributed 80 lesions across five distinct locations, resulting in 16 lesions per location (bladder anterior wall, posterior wall, left wall, right wall, and dome).

### 3.2. Technical Success Rate (TSR)

As summarized in Table 1, the TSR was significantly higher in the RBC-ERBT group than in the ERBT group (98.8% vs. 77.5%). Only one case of technical failure due to perforation was reported in the RBC-ERBT group. In the ERBT group, technical failure was noted in 18 procedures, three of which were attributed to perforation and the remaining 15 were attributed to conversion to piecemeal resection (four cases of laser piecemeal resection and 11 cases of electrocautery piecemeal resection).

Regarding lesion morphology, both exophytic and flat lesions showed significantly higher TSR in the RBC-ERBT group than in the ERBT group (97.5% vs. 77.5% and 100% vs. 77.5%, respectively). For both 1 cm and 2 cm lesion sizes, RBC-ERBT had a significantly higher TSR than did ERBT (100% vs. 85.0% and 97.5% vs. 70.0%, respectively).

The TSR of the RBC-ERBT group for dome lesions was significantly higher than that of the ERBT group (93.8% vs. 25.0%, respectively). However, for lesions located in other areas, no significant differences were noted in the TSR between the two groups.

### 3.3. Procedure Time

The procedure times for the two techniques are listed in Table 2. Overall, RBC-ERBT had significantly shorter procedure times than did ERBT (25.4 ± 8.9 min vs. 31.4 ± 13.6 min). Specifically, in flat lesions, the operating time of RBC-ERBT was significantly shorter than that of ERBT (26.8 ± 8.6 min vs. 34.3 ± 14.3 min). However, for exophytic lesions, no significant difference was noted between RBC-ERBT and ERBT.

In terms of lesion diameter, RBC-ERBT had a significantly shorter procedure time than did ERBT for lesions with a diameter of 2 cm (29.2 ± 9.2 min vs. 39.2 ± 13.4 min). However, for lesions with a diameter of 1 cm, no significant difference was noted between the two techniques.

For lesions located at the dome, the procedure time was significantly shorter for RBC-ERBT than for ERBT (37.0 ± 7.4 min vs. 47.9 ± 14.5 min). However, for lesions located in other locations, no significant difference was noted between RBC-ERBT and ERBT.

### 3.4. Safety and Quality Outcomes

Perforation was noted in one case in the RBC-ERBT group and in three cases in the ERBT group; however, the difference was not statistically significant (Table 3). In the RBC-ERBT group, all 80 lesions were resected en bloc, whereas in the ERBT group, 15 (18.8%) lesions were resected piecemeal; this difference was statistically significant. The DM sampling rate was higher in the RBC-ERBT group than in the ERBT group (98.8% vs. 96.3%); however, the difference was not statistically significant.

## 4. Discussion

Our study revealed that the RBC-ERBT group demonstrated significantly higher success rates, faster resection times, and effective management of challenging lesions compared with the ERBT group. These differences in outcomes can be attributed to the limitations of ERBT, which are often constrained by the characteristics of the lesions. However, the introduction of a traction device in RBC-ERBT allowed us to overcome these limitations and achieve improved results.

Lesion morphology and size are fundamental factors that influence the outcome of the resection procedure. In this study, we observed that the addition of a rotatable channel in RBC-ERBT resulted in higher TSRs than that in ERBT for both lesion morphologies. Regarding lesion size, although no specific cutoff size recommendation was provided for en bloc resection if specimen retrieval is feasible [3], a decrease in TSR has been reported for larger lesions [22]. Based on our experience with ERBT, we encountered difficulties in resecting larger lesions owing to the challenge of cutting along the target line, as the resected tissue tends to move and obstruct the target area. However, with the implementation of RBC-ERBT involving the use of additional forceps, we were able to lift the resected tissue, providing better visibility and tension in the target area. Through this “lift-and-cut” technique, RBC-ERBT not only demonstrated higher TSRs than ERBT for both 1 cm and 2 cm lesions but also showed reduced resection times for 2 cm lesions.

The applicability of en bloc techniques can be limited by lesion location. Lesions on the anterior and posterior bladder walls and the dome are less favorable for ERBT [3,23,24]. Laser ERBT presents a particular challenge, as the end-firing laser cannot change direction, making it more challenging to resect lesions located on the face of the endoscope (at the dome). This was due to the inability of the laser to reach the target cutting area located behind the lesion within the endoscopic field of view. Moreover, the upward angulation of the bladder wall at the dome increases the risk of inadvertent deep incisions with lasers [25]. In our study, we found that most of the failed procedures were associated with the resection of dome lesions. To address this challenge, we developed a “peel-off” method for ablating dome lesions. Based on our experience, this method enables the feasibility of RBC-ERBT at this specific location. Compared with the conventional ERBT, our RBC-ERBT with the “peel-off” method demonstrated significantly higher TSR and faster resection speed, highlighting its advantage.

In terms of the safety and quality of the resection, we believe that the use of an additional grasper for exposure and traction during laser cutting can offer improved control and precision, thereby enhancing safety and quality. In our study, we observed a higher rate of en bloc resection in the RBC-ERBT group than in the ERBT group, indicating that RBC-ERBT more often provided one-piece resection, reducing tumor fragmentation and potentially lowering the risk of reimplantation. However, the differences in the perforation rate and DM sampling rate did not reach statistical significance, suggesting further investigation may be needed to draw definitive conclusions in these areas.

We believe that the improved outcomes observed with RBC-ERBT can be attributed to the traction provided by the second rotatable channel. In the gastrointestinal field, various traction devices have been used in endoscopic submucosal dissection and shown to facilitate the surgery [26,27,28,29,30]. However, the urethra, a long, narrow, and tubular structure, poses challenges in placing traction devices into the bladder. Furthermore, since the bladder is a liquid-filled organ, applying traction through its walls may increase the risk of bladder perforation and leakage, potentially leading to tumor cell seeding outside the bladder [31].

Most transurethral resectors have rigid structures that permit rotational movements, and they typically come with a single working channel that can rotate around the camera to allow instruments access to different quadrants of the endoscopic view. Introducing a second independent rotatable channel enables adjustable traction during resection. Therefore, the bi-rotatable channel represents a practical and effective traction method for ERBT surgery.

This study has some limitations. First, anatomical differences between porcine and human bladders, such as the thinness and lack of perivesical fat, can pose challenges in achieving en bloc resection without perforation. Second, the proximity of the two ureteric orifices to the internal opening of the urethra in the porcine bladder resulted in a significantly smaller bladder trigone than that in the human bladder. Consequently, there is limited space for various types of lesions in this area. Therefore, we opted to use the posterior wall as a proxy to encompass both the posterior wall and trigone, as the techniques for resecting lesions in the bladder trigone and posterior wall are similar. Third, the artificial lesions used in our study may not have fully replicated the morphology and size variability in natural bladder tumors. It is important to acknowledge that the ex vivo bladder model cannot simulate certain aspects encountered during ERBT, such as bleeding and the obturator reflex. Therefore, further in vivo animal studies and clinical trials are required to comprehensively evaluate these factors.

## 5. Conclusions

Our study demonstrated the feasibility of RBC-ERBT in an ex vivo porcine model. Compared with conventional ERBT, RBC-ERBT showed an improved TSR for lesions of various morphologies and sizes. Reduced resection times were also achieved for flat and relatively large lesions. Notably, RBC-ERBT effectively addressed the challenges associated with difficult lesion locations, resulting in a higher TSR and faster resection speed, specifically for dome lesions. However, further research is required to assess the clinical applicability of RBC-ERBT. Overall, our findings showed that RBC-ERBT holds promise as a valuable approach for NMIBC resection, offering potential benefits to patients.

## Figures and Tables

**Figure 1 cancers-15-04255-f001:**
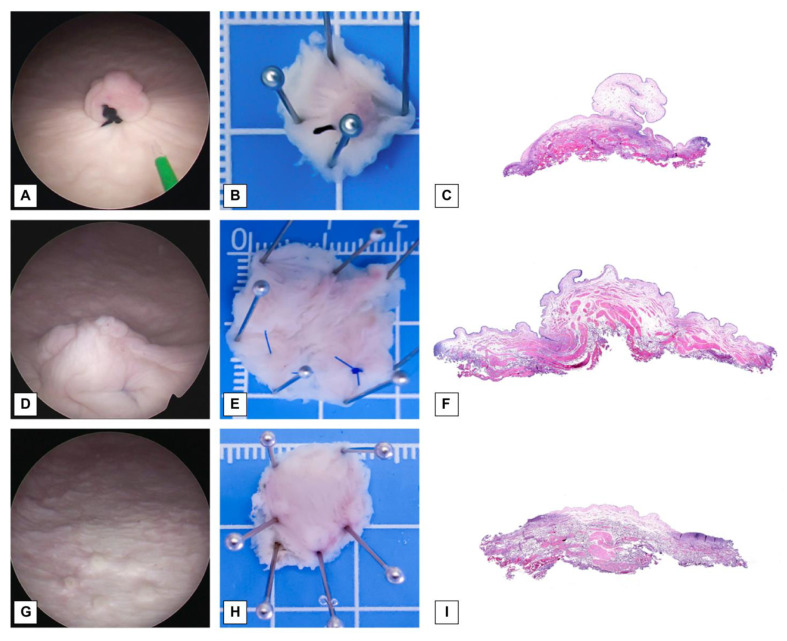
Visualization of lesions, specimen placement, and photomicrographs. (**A**) Endoscopic view of an exophytic lesion with a 1 cm diameter (original magnification × 2). (**B**) Exophytic lesion with a 1 cm diameter pinned on a plate. (**C**) Photomicrograph of an exophytic lesion with a 1 cm diameter (original magnification × 20). (**D**) Endoscopic view of exophytic lesion with a 2 cm diameter (original magnification × 2). (**E**) Exophytic lesion with a 2 cm diameter pinned on a plate. (**F**) Photomicrograph of an exophytic lesion with a 2 cm diameter (original magnification × 20). (**G**) Endoscopic view of a flat lesion with a 1 cm diameter (original magnification × 2). (**H**) Flat lesion with a 1 cm diameter pinned on plate. (**I**) Photomicrograph of a flat lesion with a 2 cm diameter (original magnification × 20). (**J**) Endoscopic view of a flat lesion with a 2 cm diameter (original magnification × 2). (**K**) Flat lesion with a 2 cm diameter pinned on plate. (**L**) Photomicrograph of a flat lesion with a 2 cm diameter (original magnification × 20).

**Figure 2 cancers-15-04255-f002:**
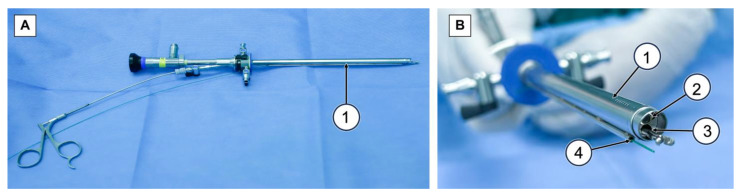
The resectoscope set for RBC-ERBT. (**A**) Side view of resectoscope set for RBC-ERBT, (1) rotatable external sheath with a second working channel. (**B**) End view of resectoscope set for RBC-ERBT, (1) rotatable external sheath, (2) endoscope, (3) internal working channel, and (4) external working channel.

**Figure 3 cancers-15-04255-f003:**
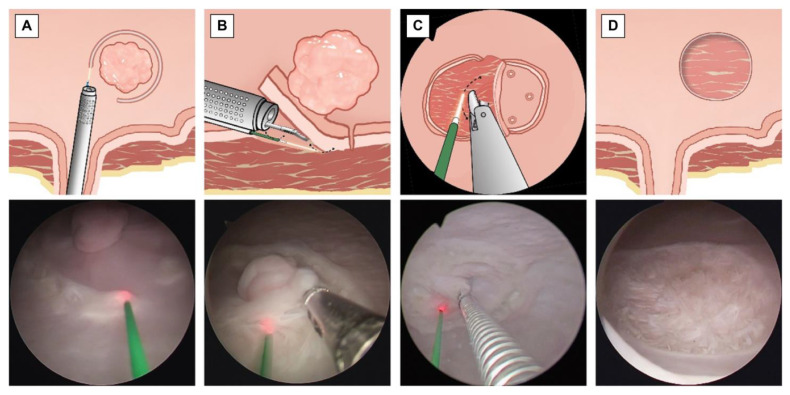
Key steps of RBC-ERBT. (**A**) Circumferential incision using a holmium laser (original magnification × 2). (**B**) “Lift-and-cut” method for lesion resection (original magnification × 2). (**C**) “Peel-off” method for lesion resection at the dome (original magnification × 2). (**D**) Changes in the bladder wall after lesion removal (original magnification × 2): The lesion bed displays visible muscle fibers without the presence of perforation.

**Figure 4 cancers-15-04255-f004:**
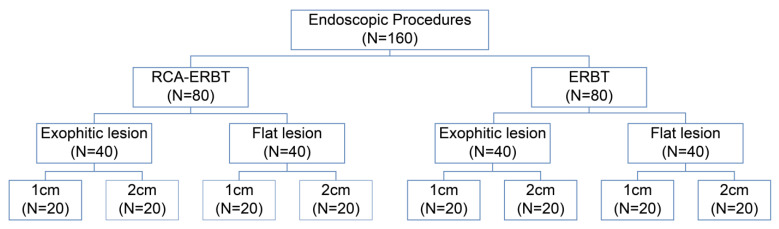
Classification of lesions included in this study.

**Table 1 cancers-15-04255-t001:** TSR results.

	RBC-ERBT		ERBT		*p*
	Number of Procedures	Technical Success, *n* (%)	Number of Procedures	Technical Success, *n* (%)	
All lesions	80	79 (98.8)	80	62 (77.5)	0.000 *
Type of lesion					
Exophytic lesion	40	39 (97.5)	40	31 (77.5)	0.007 *
Flat lesion	40	40 (100)	40	31 (77.5%)	0.005 *
Diameter of lesion					
1 cm	40	40 (100)	40	34 (85.0)	0.034 *
2 cm	40	39 (97.5)	40	28 (70.0)	0.001 *
Location of lesion					
Anterior wall	16	16 (100)	16	12 (75.0)	0.101
Posterior wall	16	16 (100)	16	14 (87.5)	0.484
Left wall	16	16 (100)	16	16 (100)	-
Right wall	16	16 (100)	16	16 (100)	-
Dome	16	15 (93.8)	16	4 (25.0)	0.000 *

* Significance at *p* < 0.05.

**Table 2 cancers-15-04255-t002:** Procedure time.

	RBC-ERBT		ERBT		*p*
	Procedures, *n*	Procedure Time, min Mean ± Standard Deviation	Procedures, *n*	Procedure Time, min Mean ± Standard Deviation	
All lesions	80	25.4 ± 8.9	80	31.4 ± 13.6	0.001 *
Type of lesion					
Exophytic lesion	40	24.1 ± 9.0	40	28.6 ± 12.2	0.066
Flat lesion	40	26.8 ± 8.6	40	34.3 ± 14.3	0.006 *
Diameter of lesion, cm					
1	40	21.7 ± 6.7	40	23.6 ± 8.3	0.244
2	40	29.2 ± 9.2	40	39.2 ± 13.4	0.000 *
Location of lesion					
Anterior wall	16	27.4 ± 7.2	16	31.1 ± 12.2	0.314
Posterior wall	16	19.9 ± 5.6	16	25.1 ± 8.7	0.057
Left wall	16	21.0 ± 6.0	16	26.4 ± 8.8	0.053
Right wall	16	21.8 ± 4.9	16	26.6 ± 8.4	0.054
Dome	16	37.0 ± 7.4	16	47.9 ± 14.5	0.012 *

* Significance at *p* < 0.05.

**Table 3 cancers-15-04255-t003:** Safety and quality outcomes.

	RBC-ERBT (*n* = 80)	ERBT (*n* = 80)	*p*
Perforation, *n* (%)	1 (1.3%)	3 (3.8%)	0.613
Piecemeal resection, *n* (%)	0 (0%)	15 (18.8%)	0.000 *
DM sampling, *n* (%)	79 (98.8%)	77 (96.3%)	0.613

* Significance at *p* < 0.05.

## Data Availability

The data analyzed in this study are available from the corresponding author upon reasonable request.

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
