# Peer review of "Rotatable Bi-Channel En Bloc Resection of Bladder Tumor for Non-Muscle-Invasive Bladder Cancer in an Ex Vivo Porcine Model"

_cancers, 2023, doi:10.3390/cancers15174255_

Round 1

Reviewer 1 Report

an interesting manuscript on a technical novelty in EBRT for bladder cancer.

Introduction - nicely written paragraph, depicting the advantages and potential pitfalls of EBRT - no remarks

Material and methods - comprehensive paragraph on the protocol of this pilot study, emphasizing all of its major benefits - specially designed double channel resectoscope, usage of all type of lesions, detailed description of the surgical technique - no remarks

Results - nicely presented and visualized - no remarks

Discussion and conclusion - in this pilot study the authors found that their novel bi-channel technique outperforms standard EBRT in efficacy (TSR and resection time), especially in dome lesions, flat lesions and lesion > 2 cm, potentially leading to significantly expanding indications for EBRT for bladder cancer 

one minor issue on row 267 - 

RBC-ERBT more often provided intact piecemeal resections - could it be a typo? should be one-piece resection maybe?

Author Response

Dear Reviewer,

We sincerely appreciate your thorough review of our manuscript and the valuable feedback you have provided. Your suggestions and opinions have been immensely helpful in refining the quality of our work. We have carefully addressed the issue you pointed out regarding the typographical error on row 267, we have corrected the error by replacing “intact piecemeal resections” with “one-piece resection.”

Regarding your suggestion to improve the methods section, we have taken your feedback into account. As part of our efforts to enhance the clarity of the RBC-ERBT, we have included images of the procedure setup for RBC-ERBT as Figure 2 in the Materials and Methods section. These images provide a visual context that we believe contribute to a better understanding of the procedure.

We would greatly value any additional suggestions or observations you may have to improve the manuscript. We look forward to working with you to move this manuscript closer to the publication of our paper in Cancers.

Best regards,

Boxin Xue

Reviewer 2 Report

It is an experimental study presenting the feasibility and efficacy of the rotatable bi-channel en bloc resection of bladder tumors in porcine models. The integration of the en bloc resection of bladder tumors and its restrictions constitute an appealing topic. 

The use of the English language is excellent.

Abstract: It summarizes successfully all significant points of each section of the study.

Introduction: It provides a background of bladder tumors and en-bloc resection. The aim of the study is clearly stated.

Material and Methods: The study is well-designed. All the available parameters regarding the evaluation of the efficacy and safety of RBC-ERBT  were recorded. The number of experiments is adequate. 

The addition of a figure of the resectoscope used with the two channels could be beneficial.

Results: The presentation of the results is very efficient. The tables and mainly the figures are very helpful and comprehensive.

Discussion: All the main points of the study were adequately discussed. The potential causes and explanations of all the observations were presented in an excellent way.

Conclusion: The conclusion is in accordance with the results.

Author Response

Dear Reviewer,

I would like to express our sincere gratitude for your thoughtful and positive review of our manuscript and the valuable insights you have imparted. Your feedback has been immensely beneficial in enhancing the quality of our work. In response to your recommendation, we have added images illustrating the RBC-ERBT surgical instruments to the Material and Methods section, providing a clearer visual understanding of the procedure setup.

We would greatly value any additional suggestions or observations you may have to improve the manuscript. We look forward to working with you to move this manuscript closer to the publication of our paper in Cancers.

Best regards,

Boxin Xue